# Efficient and Simultaneous Chitosan-Mediated Removal of 11 Mycotoxins from Palm Kernel Cake

**DOI:** 10.3390/toxins12020115

**Published:** 2020-02-12

**Authors:** Atena Abbasi Pirouz, Jinap Selamat, Shahzad Zafar Iqbal, Nik Iskandar Putra Samsudin

**Affiliations:** 1Laboratory of Food Safety and Food Integrity, Institute of Tropical Agriculture and Food Security, Universiti Putra Malaysia, 43400 UPM Serdang, Selangor, Malaysia; atenapirouz.upm@gmail.com (A.A.P.); nikiskandar@upm.edu.my (N.I.P.S.); 2Faculty of Science, Institute of Biological Science, University of Malaya, 50603 UM Kuala Lumpur, Malaysia; 3Department of Food Science, Faculty of Food Science and Technology, Universiti Putra Malaysia, 43400 UPM Serdang, Selangor, Malaysia; 4Department Applied Chemistry, Faculty of Physical Science, Government College University Faisalabad, 38000 Punjab, Pakistan; shahzad@gcuf.edu.pk

**Keywords:** chitosan, mycotoxins, detoxification, LC-MS/MS, optimization

## Abstract

Mycotoxins are an important class of pollutants that are toxic and hazardous to animal and human health. Consequently, various methods have been explored to abate their effects, among which adsorbent has found prominent application. Liquid chromatography tandem mass spectrometry (LC–MS/MS) has recently been applied for the concurrent evaluation of multiple mycotoxins. This study investigated the optimization of the simultaneous removal of mycotoxins in palm kernel cake (PKC) using chitosan. The removal of 11 mycotoxins such as aflatoxins (AFB_1_, AFB_2_, AFG_1_ and AFG_2_), ochratoxin A (OTA), zearalenone (ZEA), fumonisins (FB1 and FB2) and trichothecenes (deoxynivalenol (DON), HT-2 and T-2 toxin) from palm kernel cake (PKC) was studied. The effects of operating parameters such as pH (3–6), temperature (30–50 °C) and time (4–8 h) on the removal of the mycotoxins were investigated using response surface methodology (RSM). Response surface models obtained with *R^2^* values ranging from 0.89–0.98 fitted well with the experimental data, except for the trichothecenes. The optimum point was obtained at pH 4, 8 h and 35 °C. The maximum removal achieved with chitosan for AFB_1_, AFB_2_, AFG_1_, AFG_2_, OTA, ZEA, FB_1_ and FB_2_ under the optimized conditions were 94.35, 45.90, 82.11, 84.29, 90.03, 51.30, 90.53 and 90.18%, respectively.

## 1. Introduction

Mycotoxins constitute a large number of naturally-occurring fungal secondary metabolites with considerable toxic impacts in plant and animal products, especially in developing countries [1]. These fungi usually attack cereals and other crops on the field or in storage, thereby causing significant economic losses and affecting trade globally. Furthermore, the presence of mycotoxins in feed and food commodities is a major health concern as the toxic effects cause serious problems to the health of animals, and ultimately humans. The major toxins of concern include aflatoxins (AFB_1_, AFB_2_, AFG_1_, AFG_2_), ochratoxinA OTA, fumonisins (FB_1_, FB_2_) and trichothecenes (ZEA, HT-2, T-2, and DON), which are placed in group 1, group 2 and group 3, respectively. According to the International Agency of Research on Cancer (IARC), AFB1 has been recognized as a class 1 carcinogen for animals and humans [2]. Animals often show symptoms of mycotoxicosis even when these mycotoxins are present below permitted levels. This is probably a result of the negative effects of the synergistic interactions between various mycotoxins when they occur simultaneously in the same feed [3].

Although palm kernel cake (PKC) has been commonly used in ruminant feed, its use in poultry, swine, and fish diets is limited due to its high fiber content [4]. While PKC is a valuable source of protein and energy, it is susceptible to infection with mycotoxin-producing fungi. It is abundantly produced in mainly in the equatorial tropics, including South East Asia, Africa and South America. The incidence and severity of the growing number of reports concerning the presence of mycotoxins in feed and food highlights the heightened need for practical detoxification procedures [3,5,6]. Numerous strategies involving physical, chemical and biological methods have been explored in order to detoxify mycotoxin-contaminated food and feed materials [7]. The successful removal of mycotoxins from contaminated agricultural commodities using biological and chemical methods is hindered by several issues [8]. Some of these include safety issues, failure to effectively inactivate different mycotoxins simultaneously, limited efficacy, potential loss of nutritional quality and unfavorable cost implications. Adsorption is one of the most effective and attractive physical processes for mycotoxin removal. Its major advantage is that it has minimal to no detrimental effects on the animal and this often does not outweigh the benefits gained from the high removal efficiency. The efficacy of adsorption appears to depend on the chemical structure and interactions between the solvent, adsorbate and adsorbent [9]. Several studies have reported the applicability of some inorganic and organic adsorbents such as activated charcoal, clay, yeast cell wall, hydrated sodium calcium aluminosilicates (HSCAS) and polymers for the binding and removal of mycotoxins [6,10,11,12,13]. In these cases, single adsorbents have been shown to be effective against one or two specific mycotoxins while few adsorbents may be used for various mycotoxins. However, none of them has been effective against all toxins [3,10,14,15]. Amongst numerous adsorbents, chitosan (CTS) is an effective, eco-friendly polymer and low-cost adsorbent which is used for the removal of pollutant effluents even at low concentrations because of its excellent adsorption capability [16].

Chitosan is a polyaminosaccharide that is derived through the deacetylation of chitin and it is predominantly comprised of unbranched chains of β-(1, 4)-linked 2-deoxy-2-amino-d-glucopyranose units. Chitin and CTS are the second most abundant bio- polymers after cellulose. They can be isolated from shells of crustaceans such as crabs, krills, shrimps and cell walls of some fungi [17]. Chitosan appears to be more attractive for use as adsorbent because of its beneficial properties such as hydrophilicity, biodegradability, non-toxic nature and anti-bacterial property. Likewise, CTS is a good scavenger for pollutants binding owing to the presence of amine (–NH_2_) and hydroxyl (–OH) functional groups in its structure; these function as coordination and reaction sites [18]. In addition, CTS sorbents are conventionally used for adsorption of anionic pollutants from aqueous solutions because of their polycationic nature.

Chitin has very poor solubility in most organic solvents while CTS dissolves readily in dilute acidic solutions [19]. Aqueous acid solutions with pH below 6.0 have been found to be the best solvents for CTS. Such acids include acetic acid, hydrochloric acid, nitric acid and formic acid. However, sulfuric and phosphoric acids are not suitable solvents for CTS as it is insoluble in them [17]. Numerous articles have reviewed the application of chitosan and its derivatives in removing contaminants from water samples, wastewaters, dye compounds and heavy metals [20,21,22,23,24]. However, to date, there is limited number of literature on the successful simultaneous removal of mycotoxin with CTS as adsorbent [9,25,26].

To the best our knowledge, this is the first study on the simultaneous removal of 11 toxic mycotoxins in PKC using CTS.

## 2. Results and Discussion

### Data Analysis

Table 1 shows the experimental treatments in the CCD design of RSM applied in optimizing the removal of mycotoxins from PKC using chitosan in Table 2, the estimated regression coefficients and the corresponding *R^2^* values, *p*-value and model lack of fit are presented. Each response was evaluated as a function of main (linear), quadratic and interaction effects of pH (*x*_1_), time (*x*_2_) and temperature (*x*_3_). As revealed by ANOVA, the coefficients of multiple determination (*R^2^*) for the responses ranged between 0.89 and 0.98, thus illustrating that the quadratic polynomial models obtained adequately represented the experimental data (Table 2). The models sufficiently related the studied independent variables to the simultaneous removal of eight mycotoxins by CTS, indicating a perfect fit (*p* < 0.05) with the second-order response surface equations. A good model fit is indicated by the *R^2^* values being at least 0.80 [27] Apparently, chitosan showed poor adsorption for DON, HT-2 and T-2 as the *R^2^* values for these responses were all less than 0.80 (i.e., 0.62, 0.55 and 0.71 respectively) and their *p*-values were likewise not significant (*p* > 0.05). These low *R^2^* values indicate that the predicted responses differed considerably from the experimental values and the models were not adequate. This observation may be due to the fact that trichothecenes, being non-ionizable molecules having a bulky epoxy group, have poor adsorption with plane surfaces. As a consequence, they adsorb on very few adsorbent agents [28].

Judging from the *F* and *p* values for the main, quadratic and interaction effects of each independent variable (pH, time and temperature) as seen in Table 3, most of them showed significant effects (*p* < 0.05) for the removal of mycotoxins. Compared to the quadratic and interaction effects, the main linear effects were more significant (*p* < 0.05) for mycotoxin removal. As all factors had significant (*p* < 0.05) effects on the mycotoxin removal (Table 3), they should therefore be retained as critical parameters in the final reduced model for fitting with the experimental data. Furthermore, the interaction effects of all the independent variables were significant (*p* > 0.05) only in the removal of AFB_1_. Except in the case of ZEA, the interaction effects of pH with the other two factors was significant (*p* < 0.05) in the removal of the mycotoxins. The reason may be due to the structure and the differences in the solubility of ZEA, owing to its polarity. The resorcinol moiety of ZEA is fused to a 14-atom macrocyclic lactone ring, with a double bond in the trans isomeric form with ketone and methyl groups [29], and its deprotonated form (ZEN) exists at pH > 7.62 [30,31]. Therefore, at acidic pH, the main binding mechanism will likely be hydrophobic interactions. Therefore, the removal of ZEA depended on temperature during equilibrium time. On the other hand, the results showed that the interaction of pH and temperature was significant (*p* < 0.05) for response variables, such as AFB_1_, AFG_1_, AFG_2_, OTA and FB_1_, with the effect being most significant for AFB_1_ and OTA reduction as their *F* values were 97.42 and 70.21 respectively. The interaction effect of pH and time was significant (*p* < 0.05) for removal of AFB_1_, AFB_2_, AFG_1_ and OTA (Table 3). Likewise, the interaction of pH and time had the highest influence on the removal of AFB_1_ due to its rather high *F*-value of 127.30. Therefore, it can be observed that pH was more influential for mycotoxin removal than the other variables studied. An acidic pH causes the release of more H^+^ ions that may react with the adsorbent or adsorbate, thereby affecting results. Being a polycationic polymer, the surface of chitosan would become strongly positively charged under more acidic conditions, due to protonation of amino groups. This would cause increased electrostatic reactions between it and the negatively-charged mycotoxin molecules, leading to better adsorption. However, with gradual increase in pH, the adsorbent (chitosan) surface carries more negative charges, causing repulsion between it and the mycotoxin molecules. This eventually results in reduced adsorption capacity [20].

The interaction effects of the processing variables on mycotoxin removal, shown to be significant from ANOVA results (Table 3), are explained visually with three-dimensional (3-D) response surface plots in Figure 1a–m. It can be seen from Figure 1a–c that the removal of AFB1 was significantly affected (*p* > 0.05) by the interaction of all the independent variables. The interaction effect x_1_x_3_ was also significant (*p* < 0.05) for almost all the target mycotoxins except ZEA, AFB_2_ and FB_2_, thus indicating an overall positive effect on mycotoxin reduction. Maximum removal of AFB_1_ and AFG_2_ were seen in the middle of pH (*x*_1_) with two other factors (*x*_2_, *x*_3_) for AFB_1_ (Figure 1a–c) and pH (*x*_1_) and temperature (*x*_3_) for AFG_2_ (Figure 1g). Likewise, maximum removal of ZEA was illustrated in the middle of *x*_2_ and *x*_3_ in Figure 1j.

## 3. Statistical Design

Three independent variables, i.e., pH (*x*_1_), equilibrium time (*x*_2_) and temperature (*x*_3_), were evaluated for their effect on the response variables *y*_1_–*y*_11_, denoting the removal of DON, AB_1_, AB_2_, AG_1_, AG_2_, OTA, ZEA, HT-2, T-2 toxin, FB1 and FB2 respectively. These variables were selected for the study based on findings from literature and preliminary studies. A composite central design (CCD) with 20 experimental runs was employed to study the main and interaction effects of the variables on the mycotoxin removal. The range of values applied on the independent variable ranges studied were: pH 3–6, equilibrium time of 3–8 h and 30–50 °C temperature, all set at three levels for each variable, as illustrated in Table 4. The center point was replicated six times reproducibility [32].

## 4. Statistical Analysis

Regression analysis as well as analysis of variance (ANOVA) was applied to establish the nature of the relationship between the responses and the three independent variables. To fit the regression models to the experimental data with the objective of achieving the overall optimal region for all response variables studied [33]. For all tests the *p*-value adopted was less than 0.05. The generalized polynomial model proposed for relating the response to independent variables is given below:*y_i_* = *b_0_+b_1_x_1_+b_2_x_2_+b_3_x_3_+b_12_x_1_x_2_+b_13_x_1_x_3_+b_23_x_2_x_3_+b_11_x_12_+b_22_x_22_+b_33_x_32_*(1)
where *y*_i_ represents the predicted dependent variables; b_0_ is the offset term (constant); b_1_, b_2_ and b_3_ are the linear effects; b_11_, b_22_ and b_33_ are quadratic effects; and b_12_, b_13_, b_23_, b_31_ and b_32_ are the interaction effects. The terms *x*_i_*x*_j_ and *x*_i_^2^ (i = 1, 2 or 3) denote the interaction and quadratic terms respectively [34,35]. The adequacy of the model was tested using model analysis, lack of fit test and coefficient of determination (*R^2^*) analysis.

## 5. Optimization and Validation Procedure

The final reduced models in optimization can be presented as 3-D response surface plots. These can reveal the significant interactive effects of the independent variables on the response [36]. Here, the relationship of each response to the independent variables was expressed with 3-D plots by fixing two variables at the centre point while varying the third within the chosen experimental range. The levels of the independent variables for achieving the optimum goal of the individual and overall responses were determined with the aid of the response optimizer.

The response optimizer allows the attainment of a fair balance in the optimization of several response variables by identifying the best combination of input variable settings that favour maximum value of response(s) [37]. It was thus applied in the current study in order to simultaneously reduce the 11 target mycotoxins. The final reduced models were verified by conducting five replicate experimental runs at the optimal settings and comparing the observed results with the predicted responses. Significant difference between the predicted and experimental results were further confirmed by one sample *t*-test Experimental design, model generation, prediction and other statistical analysis were done using a statistical package (Minitab 17 software, State College, PA, USA).

## 6. Adsorption Studies

About 2 kg of fresh representative PKC samples were kept at 4 °C ahead of sample extraction and subsequent analysis. Three different concentrations (5.0, 25.0 and 100.0 ng/g) of AFB_1_, AFB_2_, AFG_1_, AFG_2_, OTA, ZEA, DON, HT-2, T-2, FB1 and FB2 standards were mixed with approximately 5 g of PKC, each in triplicate solvent evaporation was allowed to occur in the spiked samples by storing them overnight in the dark. Preliminary studies conducted within the range of 0.005 to 0.04 g of CTS revealed that decrease in mycotoxins was not observed beyond 0.035 g. Therefore, this amount was utilized in the adsorption experiments. In the sorption experiments, 350 mg of CTS adsorbent was added to 5 g of mycotoxin-contaminated PKC samples in 50 mL flasks. A 20 mL volume of solvent (acetonitrile/water/formic acid at 70:29:1, *v*/*v*/*v*) was added to the flask and the pH (3–6) was adjusted as needed using 0.01 M HCl. Adsorption was then carried out at controlled temperatures (30–50 °C) and at desired equilibrium times (4–8 h) under constant shaking (300 rpm). The mixture of solution was centrifuged at 3000 rpm for 10 min and 1 mL of the final solution was mixed with 3 mL of water for dilution [38]. The purpose of sample dilution during the sample preparation procedure was to reduce the possible matrix effect [39]. The extract obtained was then filtered with nylon syringe filter (0.22 μm). At the end of this process, residual mycotoxins present were measured using LC/MS-MS [40]. Adsorption was estimated based on the initial and final amounts of mycotoxins present in the aqueous, as presented in the following Equation (2) [41]:E= (C0−Ce)/C0/100

In Equation (2), C0 is the concentration (ng/mL) of the mycotoxin in the blank control and Ce is its concentration (ng/mL) in the supernatant.

## 7. Optimization for Maximum Mycotoxin Removal

To identify the optimum settings of the independent variables for the desired goal of mycotoxin removal, multiple response optimizations (numerical and visual) were conducted. Two stages may be considered in optimization: (a) visualize the significant interaction effects of independent variables on the response variables and (b) the actual optimization, where the factors are further examined in order to determine the best applicable conditions. Presented in Figure 2 are the respective response plots for simultaneous removal of eight mycotoxins obtained with different settings of the studied variables. Maximum removal of all mycotoxins was predicted to occur at the optimal condition of pH 4, time 8 h and temperature 35 °C (Figure 2).

## 8. Reduced Response Model Validation

Adequacy of the response-regression equations was evaluated using t-test. The corresponding experimental responses were compared with predicted values and the results are presented in Table 5. In the validation process, there must be no significant difference (*p* > 0.05) between the predicted and actual experimental values; this implies good agreement between the two values. This observation verifies adequate fitness of the response equations by RSM. Applying the optimum conditions predicted by the reduced models in this study, reduction for AFB_1_, AFB_2_, AFG_1_, AFG_2_, OTA, ZEA, FB_1_ and FB_2_ were 94.35, 45.90, 82.11, 84.29, 90.03, 51.30, 90.53 and 90.18% respectively. The total desirability was 0.77 as shown in Table 5. Hence, the final reduced models fitted by RSM were adequate.

Results of the recovery all the experimental response values showed that this method is acceptable to be used for mycotoxin removal with chitosan in PKC. Recovery values ranged from 81% to 112% for all mycotoxins as reported [38,42].

Results from this study have shown that CTS is promising as an adsorbent for removal of various types of mycotoxins. Its application for the removal of OTA in contaminated drinks has been previously demonstrated [9,43]. Dietary supplementation in poultry for removal of AFB1 and ZEA [44], demonstrated the effectiveness of CTS in reducing the levels of one or two mycotoxins. In this study, CTS showed a moderate to high adsorbent capacity against eight of the eleven mycotoxins evaluated simultaneously, though it showed poor adsorption against HT2, T-2 and DON. These findings indicate better performance against a wider range of mycotoxins than was achieved in a similar study [45] and with cross-linked chitosan [30]. This study further suggests that CTS can remarkably bind all eight of the mycotoxins assessed simultaneously.

## 9. Conclusions

We report for the first time, the simultaneous CTS adsorption of 11 mycotoxins in PKC as an ingredient of animal feed. Statistical optimization using RSM (CCD) was a valuable tool for maximizing the effects and interactions of pH, time and temperature for removal of mycotoxins. The optimum condition for the removal of mycotoxins was at pH 4 for 8 h and at temperature of 35 °C, with high overall desirability (D 0.77). The overall coefficient of determination values for the regression models were high (0.89 < *R^2^* < 0.98), as revealed by ANOVA. Results from the current study revealed the removal of eight target mycotoxins using Chitosan. Mycotoxin removal efficiency was 45.90% and 94.35% for AFB2 and AFB1 respectively. In increasing order, removal efficiency generally followed this trend: AFB_1_ > FB_1_ > FB_2_ > OTA > AFG_2_ > AFG_1_ > ZEA and AFB_2_. The present method offers clear advantages in terms of simplicity, speed, cost-effectiveness and ensuring low concentration of adverse mycotoxins in the sample. This finding is important as the mycotoxins were diminished to a greater extent than was reported using other adsorbents. The results clearly showed that pH is an important primary factor to be considered in the removal of mycotoxins with CTS. Lastly, CTS is relatively inexpensive and is thus a good candidate for practical applications involving simultaneous removal of mycotoxins in animal feed.

## 10. Materials and Methods

Palm kernel cake (PKC) samples were collected from local mills across different regions in Malaysia (Shah Alam and Kelantan in Selangor and Kelantan state respectively). Representative samples of the PKC were prepared as described previously [46]. Analytical pure standards of the aflatoxins (AFB_1_, AFB_2_, AFG_1_, and AFG_2_), ochratoxin A (OTA) zearalenone (ZEA), trichothecenes (deoxynivalenol (DON), HT-2 and T-2 toxin) and fumonisins (FB_1_-FB_2_), were purchased from VICAM (Watertown, MA, USA). Chitosan (CTS > 85% deacetylation) was sourced from Sigma-Aldrich (St. Louis, MO, USA). Deionized water was prepared with a water purifier (Elga Classic UV MK2; Elga, Marlow, UK). HPLC-grade solvents (acetonitrile, methanol and formic acid) were from Merck (Darmstadt, Germany). Filtration of all eluents was done using 0.22-μm Whatman membrane filters (Whatman, 110 Maidstone, UK).

### Mycotoxins Analysis by LC–MS/MS

The mass spectrometer used for the analyses was an Agilent 1290 Infinity UHPLC module LC/MS-MS with a Triple Quad LC/MS (Agilent 6410, Agilent technologies, Palo Alto, CA, USA). This system consisted of an auto sampler, a degasser and column oven. Separation was performed using a Zorbax Eclipse plus C18 column (2 × 150 mm, 3 μm) at column temperature of 30 °C manufactured by Agilent Technologies (Palo Alto, CA, USA). The analysis was operated in positive and negative modes with electrospray interface (ESI±) with the following parameters: capillary voltage of 4 kV, nitrogen as spray gas and desolvation temperature 40 °C. Mycotoxins were analyzed in multi reaction monitoring (MRM) mode while matrix-matched standard calibration was used for quantification. As shown in Table 6, the mobile phase consisted of a gradient of 2 solvents: mobile phase A (methanol) slightly acidified with mobile phase B (0.1% formic acid in water), at a flow-rate of 0.2 mL/min. Validation of the LC-MS/MS method was carried out by investigating the basic performance characteristics included linearity, limit of detection, limit of quantification and recovery in accordance with the European Commission regulation for the performance of analytical methods (EC 657/2002).

## Figures and Tables

**Figure 1 toxins-12-00115-f001:**
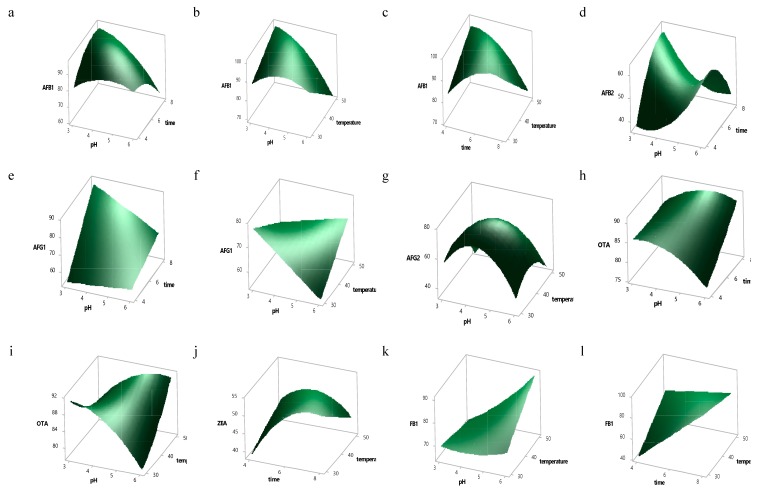
Response surface plots showing the interaction effect of independent variables on the reduction of 8 mycotoxins (**a**–**c**) AFB_1_, (**d**) AFB_2_, (**e**–**f**) AFG_1_, (**g**) AFG_2_, (**h**–**i**) OTA), (**j**) ZEA), (**k**–**l**) FB_1_, (**m**) FB_2_.

**Figure 2 toxins-12-00115-f002:**
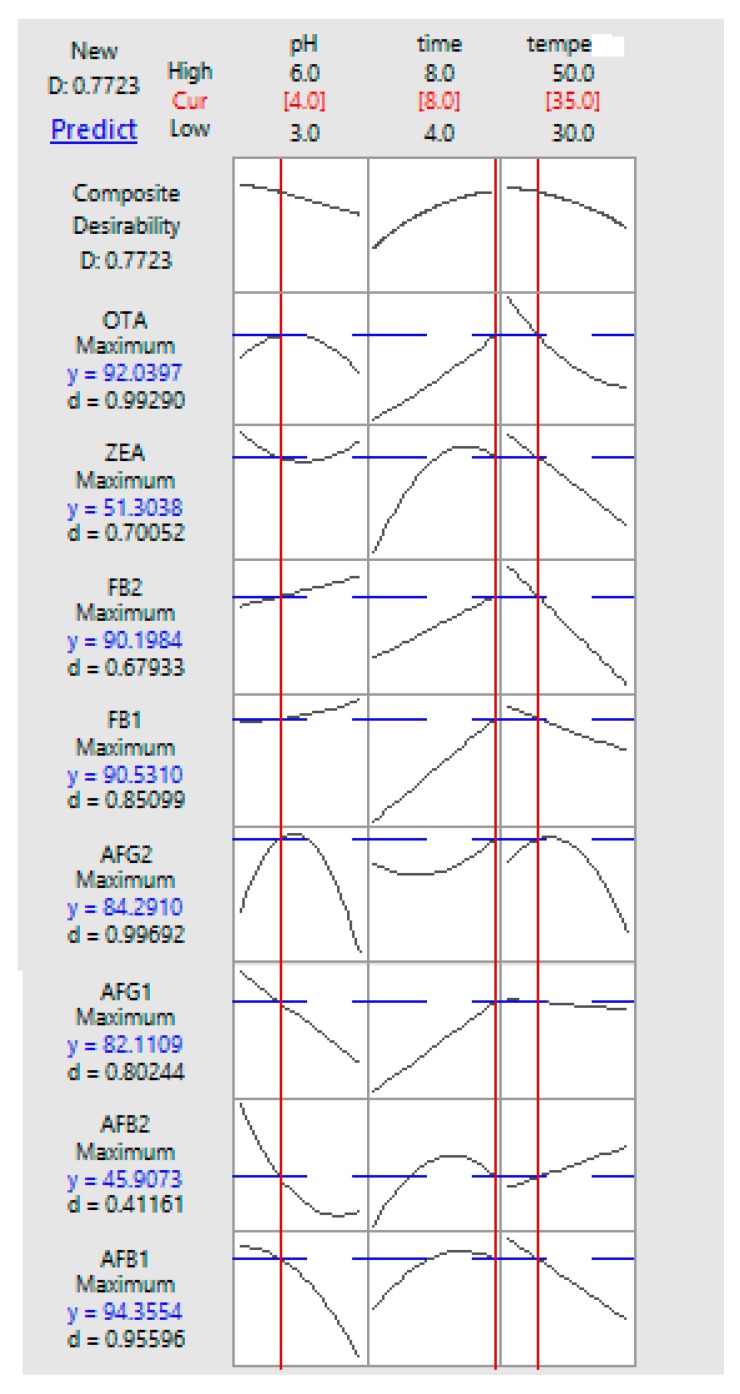
Response optimization, parameters, predicted response (*y*) and desirability of multi-mycotoxin by CTS.

**Table 1 toxins-12-00115-t001:** Experimental design (coded) of the central composite design (CCD).

Std Order	Block	Run Order	Pt Type	pH	Time (h)	Temperature (°C)
20	1	0	3	4.5	6	40
16	2	−1	3	4.5	8	40
17	3	−1	3	4.5	6	30
18	4	−1	3	4.5	6	50
19	5	0	3	4.5	6	40
13	6	−1	3	3.0	6	40
14	7	−1	3	6.0	6	40
15	8	−1	3	4.5	4	40
10	9	1	2	6.0	8	50
9	10	1	2	3.0	4	50
11	11	0	2	4.5	6	40
12	12	0	2	4.5	6	40
8	13	1	2	3.0	8	30
7	14	1	2	6.0	4	30
2	15	1	1	6.0	8	30
4	16	1	1	3.0	8	50
5	17	0	1	4.5	6	40
1	18	1	1	3.0	4	30
6	19	0	1	4.5	6	40
3	20	1	1	6.0	4	50

**Table 2 toxins-12-00115-t002:** Regression coefficient, *R^2^*, *p*-value and lack of fit test for the reduced response surface models.

Regression Coefficient	DON	AFB_1_	AFB_2_	AFG_1_	AFG_2_	OTA	ZEA	HT-2	T-2	FB_1_	FB_2_
b_0_	-	93.4	50.7	67.0	76.4	86.68	50.37	−53.0	-	72.2	81.8
b_1_	-	−6.6	1.1	−3.9	−3.3	−0.20	−0.25	12.4	-	7.2	2.9
b_2_	-	−2.5	0.88	22.1	2.7	1.9	3.3	-		15.8	3.5
b_3_	-	−0.5	3.9	2.07	−7.0	0.23	−1.1	-	-	3.1	−6.7
b_12_	-	−6.8	8.00	-	−21.8	−1.7	1.9	−31.9	-	2.2	-
b_22_	-	−1.4	−5.8	-	5.3	1.6	−4.3	-	--	-	-
b_32_	-	−6.72	-	-	−12.2	3.14	-	-	-		-
b_12_	-	−9.5	−11.2	−11.9	-	−0.7	-	-	-	-	-
b_13_	-	−8.3	-	15.7	3.3	1.4	-	-	-	5.5	-
b_23_	0.9	−9.2	-	-	-	-	−3.9	-	-	−10.5	−5.0
*R^2^*	0.72	0.98	0.97	0.96	0.98	0.98	0.89	0.61	0.71	0.98	0.92
*R^2^* (adj.)	0.53	0.96	0.95	0.94	0.97	0.96	0.82	0.47	0.38	0.97	0.89
Regression (*p*-value)	0.10 *	0.00	0.00	0.00	0.01	0.00	0.00	0.1 *	0.3 *	0.00	0.00
Lack of fit (*F*-value)	1.78	5.52	5.51	27.84	3.46	0.79	6.05	1.57	0.02	6.08	0.7
Lack of fit (*p*-value)	0.33 *	0.09 *	0.09 *	0.1 *	0.16 *	0.63 *	0.08 *	0.39 *	1.0 *	0.08 *	0.60 *

* Non-Significant (*p* > 0.05).

**Table 3 toxins-12-00115-t003:** Significant Probability (*p*-value and *F*-value) of the independent variable effects in the reduced response surface models.

Variables	*p* and F Value	Linear Effects			Quadratic Effects			Interaction Effects		
		*x* _1_	*x* _2_	*x* _3_	*x* _12_	*x* _22_	*x* _32_	*x* _1_ *x* _2_	*x* _1_ *x* _3_	*x* _2_ *x* _3_
AFB1(*y*_1_)	*p*-value	0.00	0.009	0.011	0.002	0.002	-	0.00	0.000	0.00
	*F*-value	79.66	11.21	0.82	19.44	18.47	-	127.30	97.42	120.34
AFB2(*y*_2_)	*p*-value	0.08a	0.15a	0.00	0.000	0.00	-	0.000	-	-
	*F*-value	3.72	2.40	46.57	59.37	59.84	-	304.58	-	-
AFG1(*y*_3_)	*p*-value	0.001	0.000	0.26a	-	-	-	0.00	0.00	-
	*F*-value	25.90	203.65	1.94	-	-	-	45.96	74.78	-
AFG2(*y*_4_)	*p*-value	0.001	0.004	0.00	0.00	0.00	0.00	-	0.002	-
	*F*-value	−3.33	2.72	−7.00	−21.94	5.09	−12.38	-	14.60	-
OTA(*y*_5_)	*p*-value	0.21a	0.00	0.15a	0.00	0.00	0.00	0.002	0.00	-
	*F*-value	1.90a	180.22	2.44a	33.64	32.26	119.93	18.84	79.21	-
ZEA(*y*_6_)	*p*-value	0.66a	0.00	0.06	0.03	0.001	-	-	-	0.003
	*F*-value	0.2	34.04	4.10	6.20	19.61	-	-	-	33.46
FB1(*y*_7_)	*p*-value	0.00	0.00	0.001	-	-	-	-	0.00	0.00
	*F*-value	98.15	483.12	19.42	-	-	-	-	45.99	171.58
FB2(*y*_8_)	*p*-value	0.000	0.005	0.007	-	-	-	-	-	0.00
	*F*-value	13.69	20.69	74.65	-	-	-	-	-	40.01

^a^ Non-Significant (*p* > 0.05).

**Table 4 toxins-12-00115-t004:** Levels of experimental variables established in accordance with central composite design (CCD).

Independent Variable	Independent Variable Level		
	Low	Center	High
pH	3	5	6
Time (h)	4	5	8
Temperature (°C)	30	40	50

**Table 5 toxins-12-00115-t005:** Comparison between predicted and experimental values based on the final reduced model.

Response	pH	Time	Temperature	*y* _0_	*y* _i_	*y*_0_–*y*_i_	Desirability
AFB_1_	4	8	35	94.35 ± 1.94	92.95 ± 2.1	1.4	0.95
AFB_2_	4	8	35	45.90 ± 0.003	46.58 ± 0.05	−0.68	0.41
AFG_1_	4	8	35	82.11 ± 0.84	79.48 ± 0.08	−2.63	0.81
AFG_2_	4	8	35	84.29 ± 0.31	83.11 ± 0.43	−1.18	0.99
OTA	4	8	35	90.03 ± 0.5	87.96 ± 0.27	−2.07	0.99
ZEA	4	8	35	51.30 ± 0.21	52.61 ± 0.05	2.31	0.70
FB_1_	4	8	35	90.53 ± 0.43	89.85 ± 0.52	−0.68	0.85
FB_2_	4	8	35	90.18 ± 2.3	88.73 ± 0.12	−1.45	0.68

*y*_0_: predicted value, *y*_i_: experimental value, *y*_0_–*y*_i_: residue.

**Table 6 toxins-12-00115-t006:** Gradient elution program of the LC/MS-MS.

Step	Time (min)	Solvent A%	Solvent B%	Flow Rate (mL/min)
1	0–8	10	90	0.4
2	8–10	90	10	0.4
3	10–17	0	100	0.4
4	17–20	90	10	0.4

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
