# Peer review of "Efficient and Simultaneous Chitosan-Mediated Removal of 11 Mycotoxins from Palm Kernel Cake"

_toxins, 2020, doi:10.3390/toxins12020115_

Round 1

Reviewer 1 Report

The MS "Efficient and simultaneous chitosan-mediated removal of 11 mycotoxins from palm kernel cake" give interesting and exhaustive results on the possible use of Chitosan as mycotoxin remover from conventional feedingstuff. The results are well discussed giving an exhaustive and detailed statistical approach for the aim of the work.

However, the present work needs improvements regarding the method of investigation used and a deep comparison with other works reported in literature. All my suggestions are listed below:

Introduction

Page 2 Line 43-44: Give citation on this

Page 2 Lines 75-78: This is not appropriate for the introduction section. I suggest to add the aim of the work before the M & M section

Materials and Methods

Line 86: AA must give information on the purity grade of the solvents used

Line 91: Give all the 3 concentrations reported

Given the extremely importance of AFs detection in this work after chitosan treatment, I think is important to test the trueness of the LC-MS/MS method by the study of recovery according to the EC Decision 657/2002.

Line 129: Give information on the p-value adopted.

Results and Discussion

AA cited the work of Zhiyong Zhao et al. (2015) on the synthesis of three types of cross-linked chitosan polymers and further investigation on their AFs adsorption capability. AA must compare their results with the results of this work considering that they used Chitosan concentration as parameter to be evaluated for the CCD analysis in contrast to what was described in this work (AA add the temperature). I think it could be interesting to reformulate the statistical analysis by adding 4 experimental variables for the CCD.

Author Response

Efficient and simultaneous chitosan-mediated removal of 11 mycotoxins
from palm kernel cake

Response to Reviewer 1 Comments

Point 1: Page 2 Line 43-44: Give citation on this

Response 1: Respective reference has been added and highlighted.

Point 2: Page 2 Lines 75-78: This is not appropriate for the introduction section. I suggest to add the aim of the work before the M & M section.

Response 2: We thank the reviewer for this suggestion. Correction has been revised and highlighted.

Point 3: Line 86: AA must give information on the purity grade of the solvents used

Response 3: Thank you for pointing this out. Correction has been added and highlighted

Point 4: Line 91: Give all the 3 concentrations reported

Response 4:  Concentration has been added and highlighted in Adsorption studies.

Point 5: Given the extremely importance of AFs detection in this work after chitosan treatment, I think is important to test the trueness of the LC-MS/MS method by the study of recovery according to the EC Decision 657/2002.

Response 5: Thank you for highlighting this important point. In fact, we have carried out LOD, LOQ and recovery for validation our method. In the current study, we did not bring this result due to word count. We have now added respective information in both the Mycotoxins analysis by LC–MS/MS and Reduced response model validation regarding the validation of method.

Point 6: Line 129: Give information on the p-value adopted.

Response 6: As recommended, the p-value adopted have been added and highlighted.

Point 7: AA cited the work of Zhiyong Zhao et al. (2015) on the synthesis of three types of cross-linked chitosan polymers and further investigation on their AFs adsorption capability. AA must compare their results with the results of this work considering that they used Chitosan concentration as parameter to be evaluated for the CCD analysis in contrast to what was described in this work (AA add the temperature). I think it could be interesting to reformulate the statistical analysis by adding 4 experimental variables for the CCD.

Response 7: We thank the reviewer for this suggestion. While the authors agree with the reviewer that there is a little difference between Zhao et al. (2015) and our work, but because of lack of knowledge regarding the mycotoxin removal simultaneously with chitosan, we have been compared. Study on the effect of 4 experimental variables such as pH, time, temperature and concentration of chitosan would be great idea and need to be considered in the near future.

Reviewer 2 Report

The manuscript describes the removal of 11 mycotoxins commonly present in food and feed by using chitosan as an adsorbent in palm kernel cake samples. The design of the study is well planned and the different assays carried out are adequate to obtain what the authors are looking for. However, in my opinion, results obtained are not well described and some misunderstood can occur when trying to interpret them. In addition some contradictory information is presented in different paragraphs along the manuscript and some methods employed and presented as results are not completely described in materials and methods section.

The authors have to take into consideration some remarks listed below:

Line 12: the unit h (hour) is for time, not pH.

Line 13: 4-8 minutes? Hours?

Line 29: these mycotoxins can be abbreviated and then the abbreviated form can be used in next paragraphs.

Line 30: (IARC)

Line 31: Here, the IARC document where is set the carcinogenicity of these mycotoxins must be cited.

Line 81-82: “Representative samples of the PKC were prepared as described previously (reference nature) previously”. Previously, where? What does reference nature mean?

Line 91: Three different concentrations (5.0 - 100.0 ng/g)… There are only 2 values.

Line 96: why extracts are diluted? Did the authors detect an important matrix effect?

Line 98: samples are injected in the LC-MS/MS to remove mycotoxins???

Line 99: 2.2. Mycotoxins analysis with LC–MS/MS àmycotoxin analysis by LC-MS/MS

Line 100: The mass spectrophotometer… Is a mass spectrometer, not spectrophotometer!

Line 103: 30 ºC à 30 °C

Line 106: 40 ºC à 40 °C.  This is repeated throughout the manuscript.

Line 107: “matrix-matched standard calibration was used for quantification”. This is due to matrix effect? Is this the reason why samples are diluted? In my opinion this fact must be described in the manuscript.

Line 111: comprising and an auto sampler à comprising an auto sampler…

Line 113: “Electrospray ionisation (ESI) was used for the operation of the MS system in negative and positive ion modes”. In line 104 it is said that the analysis have been carried out in positive mode. This is contradictory.

Lines 111-117: The specified parameters are not in accordance with the previous paragraph. Other parameters and other column are described in this paragraph.

Table 2: the units must be indicated as a footnote, for example (°C, h).

Line 159: Why methanol-water mixture? In section “sample preparation” it is said that the solvent is acetonitrile/water/formic acid at 70:29:1, v/v/v.

Line 180: R2 or R2 ? Sometimes is used the first form and sometimes the second one.

Table 4 is difficult to understand. In addition, the format must be improved. What does * mean?

Line 198: and the pH?

Author Response

Efficient and simultaneous chitosan-mediated removal of 11 mycotoxins
from palm kernel cake

Response to Reviewer 2 Comments

Point 1: Line 12: the unit h (hour) is for time, not pH.

Response 1: Correction has been made and highlighted.

Point 2: Line 13: 4-8 minutes? Hours?

Response 2: Correction has been added and highlighted.

Point 3: Line 29: these mycotoxins can be abbreviated and then the abbreviated form can be used in next paragraphs.

Response 3: Correction has been made and highlighted.

Point 4: Line 30: (IARC)

Response 4: Correction has been added and highlighted.

Point 5: Line 31: Here, the IARC document where is set the carcinogenicity of these mycotoxins must be cited.

Response 5: Correction has been revised and respective reference has been added.

Point 6: Line 81-82: “Representative samples of the PKC were prepared as described previously (reference nature) previously”. Previously, where? What does reference nature mean?

Response 6: Respective reference has been added and highlighted.

Point 7: Line 91: Three different concentrations (5.0 - 100.0 ng/g). There are only 2 values.

Response 7: The second concentration has been added in Section 6 (Adsorption Studies) and highlighted.

Point 8: Line 96: why extracts are diluted? Did the authors detect an important matrix effect?

Response 8: Dear reviewer, thank you for pointing this out.  the purpose of sample dilution during the sample preparation procedure was to reduce the possible matrix effect. In fact, we have evaluated the matrix effect. The corrections for the matrix effects were performed using two calibration approaches: external matrix-matched calibration and internal standard calibration. Matrix-matched calibration was ultimately used for accurate quantification, and the recoveries. In the current study, we did not bring this result due to word count.

Point 9: Line 98: samples are injected in the LC-MS/MS to remove mycotoxins???

Response 9: Correction has been revised and highlighted in section 6 (Adsorption studies).

Point 10: Line 99: 2.2. Mycotoxins analysis with LC–MS/MS àmycotoxin analysis by LC-MS/MS

Response 10: Thank you to the reviewer for pointing this out we have revised the word and highlighted.

Point 11: Line 100: The mass spectrophotometer. Is a mass spectrometer, not spectrophotometer!

Response 11: Correction has been revised and highlighted.

Point 12: Line 103: 30 ºC à 30 °C

Response 12: Correction has been made and highlighted.

Point 13: Line 106: 40 ºC à 40 °C.  This is repeated throughout the manuscript.

Response 13: Correction has been made throughout the manuscript.

Point 14: Line 107: “matrix-matched standard calibration was used for quantification”. This is due to matrix effect? Is this the reason why samples are diluted? In my opinion this fact must be described in the manuscript.

Response 14: Thank you for pointing this out. We agree with this comment. Therefore, we have been added respective information in Section 6 (Adsorption studies).

Point 15: Line 111: comprising and an auto sampler à comprising an auto sampler

Response 15: Correction has been made and highlighted.

Point 16: Line 113: “Electrospray ionisation (ESI) was used for the operation of the MS system in negative and positive ion modes”. In line 104 it is said that the analysis have been carried out in positive mode. This is contradictory.

Response 16: Correction has been revised and highlighted.

Point 17: Lines 111-117: The specified parameters are not in accordance with the previous paragraph. Other parameters and another column are described in this paragraph.

Response 17: We thank the reviewer for these observations. In this revision we have been revised this paragraph.

Point 18: Table 2: the units must be indicated as a footnote, for example (°C, h).

Response 18: Correction has been made and highlighted.

Point 19: Line 159: Why methanol-water mixture? In section “sample preparation” it is said that the solvent is acetonitrile/water/formic acid at 70:29:1, v/v/v.

Response 19: We strongly appreciate the reviewer comment on this point. We are sorry to made a mistake, which we mentioned in the manuscript that just methanol and water has been used for extraction whereas after careful consideration and recheck the methods (Jinap Selamat et al., 2014)1 the solvent we used is a mixture of acetonitrile/water/formic acid at 70:29:1, v/v/v. Correction has been revised and highlighted.

Point 20: Line 180: R2 or R2 ? Sometimes is used the first form and sometimes the second one.

Response 20: Correction has been made throughout the manuscript.

Point 21: Table 4 is difficult to understand. In addition, the format must be improved. What does * mean?

Response 21: Correction Was done according to the comments and highlighted.

Point 22: Line 198: and the pH?

Response 22: Dear reviewer, this comment is not clear enough.

Reference

Yibadatihan, S.; Jinap, S.; Mahyudin, N. J. F. A.; A, C. P. Simultaneous determination of multi-mycotoxins in palm kernel cake (PKC) using liquid chromatography-tandem mass spectrometry (LC-MS/MS). 2014, 31, 2014.

Round 2

Reviewer 1 Report

AA answer satisfactorily to all my suggestion and comments. The paper could be accepted as it is.

Reviewer 2 Report

I would like to appreciate the effort made by the authors to correct all the suggestions; in my opinion, the manuscript is now acceptable for publication.